# Mycotoxins in Ethiopia: A Review on Prevalence, Economic and Health Impacts

**DOI:** 10.3390/toxins12100648

**Published:** 2020-10-08

**Authors:** Firew Tafesse Mamo, Birhan Addisie Abate, Kassahun Tesfaye, Chengrong Nie, Gang Wang, Yang Liu

**Affiliations:** 1School of Food Science and Engineering, Foshan University, Foshan 528231, China; f.tafesse80@gmail.com (F.T.M.); niecr@126.com (C.N.); 2Key Laboratory of Agro-Products Quality and Safety Control in Storage and Transport Process, Ministry of Agriculture and Rural Affairs, Institute of Food Science and Technology, Chinese Academy of Agricultural Sciences, Beijing 100193, China; gang0805@126.com; 3Bahir Dar Institute of Technology, Bahir Dar University, Bahir Dar 79, Ethiopia; 4Ethiopian Biotechnology Institute, Addis Ababa 5954, Ethiopia; birhanaddisie@gmail.com (B.A.A.); kassahun.tesfaye@aau.edu.et (K.T.)

**Keywords:** mycotoxins, aflatoxins, fumonisins, ochratoxins, trichotecenes, standards, hepatocellular carcinoma, Ethiopia

## Abstract

Mycotoxigenic fungi and their toxins are a global concern, causing huge economic and health impacts in developing countries such as Ethiopia, where the mycotoxin control system is inadequate. This work aimed to review the occurrences of agriculturally essential fungi such as *Aspergillus*, *Fusarium*, and *Penicillium* and their major mycotoxins in Ethiopian food/feedstuffs. The incidents of crucial toxins, including aflatoxins (B1, B2, G1, G2, M1), fumonisins (B1, B2), zearalenone, deoxynivalenol, and ochratoxin A, were studied. The impacts of chronic aflatoxin exposure on liver cancer risks, synergy with chronic hepatitis B infection, and possible links with Ethiopian childhood malnutrition were thoroughly examined. In addition, health risks of other potential mycotoxin exposure are also discussed, and the impacts of unsafe level of mycotoxin contaminations on economically essential export products and livestock productions were assessed. Feasible mycotoxin mitigation strategies such as biocontrol methods and binding agents (bentonite) were recommended because they are relatively cheap for low-income farmers and widely available in Ethiopia, respectively. Moreover, Ethiopian mycotoxin regulations, storage practice, adulteration practice, mycotoxin tests, and knowledge gaps among value chain actors were highlighted. Finally, sustained public awareness was suggested, along with technical and human capacity developments in the food control sector.

## 1. Introduction

Fungi, commonly called molds, produce various secondary metabolites known as mycotoxins [1]. They can colonize diversified food/feed commodities and produce mycotoxins during the pre-harvest stage in the field or post-harvest stage in the production chain [2,3]. Mycotoxins such as aflatoxins (AFs), fumonisins (FUMs), deoxynivalenol (DON), ochratoxin A (OTA), and zearalenone (ZEN) are agriculturally important [4], and dietary mycotoxins exposures are associated with many chronic health risks, such as cancer [5], immune suppression, and digestive, blood, and nerve defects [6]. Some mycotoxins, such as AFs, are known to cause growth impairments in children [7]. Animal feed contaminated with significant amount of mycotoxins have also been reported to cause several animal health implications such as feed refusal, vomiting, weight loss, reproduction defects, and other severe impacts [6,8]. Therefore, both food/feed loss owing to mycotoxin contamination and related hazardous health risks are a global concern [9]. The Food and Agriculture Organization (FAO) has estimated that about one-fourth of global crops are affected by mycotoxins [10], resulting in loss of billions of dollars annually [8,11]. The overall impact of mycotoxins is significant in developing countries such as Africa, particularly in sub-Saharan African region because agriculture is a crucial sector in this part of the world.

The sub-Saharan African region is highly dependent on rain-fed agriculture with majority (80%) of smallholding farmers practicing a crop livestock mixed farming system. Countries in this region are highly dependent on agriculture for food security, export market, and rural livelihoods. On average, the agriculture sector contributes to 15% of the total GDP (in some countries, the contribution of agriculture sector is >50%) and provides a living for a significant fraction (>50%) of its population [12]. However, the existing subsistence farming system in this region is vulnerable to mycotoxins contamination.

Several studies conducted in Africa have reported higher mycotoxin contamination of commodities, including maize, groundnuts, sorghum, sweet potato, and spices [9,13,14]. As a result, the health implications owing to mycotoxin diet exposure are also significantly higher. According to Liu et al. [15], about 40% of liver cancer incidences in Africa have been attributed to dietary AFs exposure. Mycotoxins contamination has severely affected Africa resulting in huge economic loss; for instance, AFs contamination of crops alone has been reported to cause an annual loss of more than USD 750 million [9]. 

The major factors that contribute to the significant impact of mycotoxins in Africa have been identified as climate change [16], lack of awareness, poor agricultural practices, and pre- and post-harvest management [13]. In addition, unavailability of modern infrastructures such as drying and storage facilities [17] as well as inadequate marketing and transportation systems have intensified the effects of these factors. Besides, the absence of proper legislations coupled with inadequate regulations, poor surveillance system, and lack of implementation capacity [9] also contribute to widespread mycotoxins contamination across several crops of Africa.

Ethiopia is one of the sub-Saharan African countries with a population of over 100 million. Approximately 85% of Ethiopia’s population is rural-based, and the country’s economy is dominated by agriculture, involving 72% of the total population. Between 2002 and 2017, agriculture accounted for more than 40% of the GDP [12] and formed more than 70% of the country’s export market [18]. However, about 74% of Ethiopian farmers are smallholders [19,20,21], and hence, the use of modern harvesting techniques, transport, storage, and application of improved fertilizer and seed varieties are limited, consequently facilitating mycotoxins contamination [11].

In Ethiopia, studies on mycotoxins have been conducted for more than four decades, with the earliest AFs study reported in 1981 [22]. Since then, a number of researchers have indicated the prevalence of different mycotoxins in diverse food and feed commodities [23,24,25]. Recently, Ayelign and Saeger [26] reviewed the current status of mycotoxins and its implications to food safety, and summarized the prevalence of different mycotoxins across different food items. However, little attention had been paid to molds responsible for the production of mycotoxins.

The growth of mycotoxigenic molds in major crops in Ethiopia and its correlations to the country’s vast agro-ecological disparity, farmers’ poor storage practices, and adulteration practices have never been documented. The economic losses incurred to date owing to the rejection of the country’s economically essential export products in response to unsafe levels of mycotoxins and mycotoxins’ impacts on the vast livestock resources should be recognized. However, the link between AFs exposure and the country’s higher childhood malnutrition rate, hepatitis B virus (HBV) infection, and high incidences of liver cancer has not been documented, and the possible health risks owing to other mycotoxins exposures must be determined. Therefore, this work aims to comprehensively review the occurrences of both mycotoxigenic fungi and their metabolites in food/feedstuffs. The impacts of mycotoxins on public health and the economy are discussed, mycotoxin tests and knowledge gaps among value chain actors in Ethiopia are highlighted, and feasible mitigation strategies along with future perspectives are provided.

## 2. Mycotoxigenic Fungi and Growth Conditions

Fungi can grow and reproduce almost everywhere in the world under variable climatic conditions ranging from arid to tropical moist and temperate environment [27]. Soil is the primary reservoir for several fungal species, including mycotoxin-producing fungi [27]. Mycotoxin-producing fungi usually belong to one of the three genera—*Aspergillus*, *Penicillium*, or *Fusarium* [28]. These fungi can invade and spoil a wide variety of agricultural products in the field and beyond. Fungal growth and mycotoxin production depend on essential factors, such as climate [16], temperature, humidity, and water activity (aw) [29]. The growth of fungi usually occurs at temperatures of 10 °C–40 °C and aw of above 0.70 [8]. Besides, higher grain moisture (16–30%) and warm grain temperature (25 °C–32 °C) favor fungal growth and mycotoxins production in stored grains [11]. 

Many species in the genera *Penicillium* and *Aspergillus* can be described as moderate xerophiles.Xerophilic fungi are characterized as being capable of growing under conditions of reduced aw (<0.85), and are most commonly associated with dried foodstuffs like cereals, nuts, and spices [27]. Some *Aspergillus* such as *Aspergillus restrictus* and *Aspergillus penicillioides*, are even more xerophilic, they grows very slowly or not at all at high aw (>0.93) and are capable of growth down to at least 0.78 aw. Whereas, many penicillia can grow between 0.85 and 0.80 aw, but they compete poorly at these low aw levels unless other factors are in their favour. Still, a few *Penicillium* species such as *P. brevicompactum* and *P.implicatum* are capable of growth below 0.80 aw [30]. Although both of these fungal genera can grow well in a temperature ranges of 25 °C–33 °C, *Aspergillus* spp. can adapt to higher temperatures (30 °C–40 °C) and higher relative humidity (>80%) [31].

Fungi such as *A. flavus* and *A. parasiticus* are essential members of the genus *Aspergillus*, which produce different kinds of AFs, such as AFB1, B2, G1, and G2 [32], as well as AFM1 [33], which is the hydroxylated metabolite of AFB1 and is found in milk and dairy products obtained from livestock that have ingested contaminated feed. Both *A. flavus* and *A. parasiticus* are most frequently detected in agricultural products because of their widespread distribution [34]. Some *Aspergillus* spp., mainly *A. ochraceus*, *A. niger,* and *A. carbonarius,* can produce different kinds of ochratoxins, with OTA known to be more toxic and the most frequently detected [35].

Some field fungi such as *Fusarium* spp. require higher relative humidity (70–90%) [31], temperatures of 20 °C–30 °C, and aw of 0.97–0.995 for their effective growth and mycotoxins production [36]. However, these fungi could also generate mycotoxins even at lower temperatures close to 0 °C, without significant fungal proliferation [8]. *Fusarium* is one of the main plant pathogenic mold genera widely distributed around the world, which causes a wide range of plant diseases in tropical and moderate climate zones [37]. Fusarium head blight is the primary disease in cereals, which is mainly caused by *Fusarium graminearum* and can significantly reduce the yield and grain quality; this disease has been reported by several researchers in Ethiopia [38,39]. Infections of crops by *Fusarium* spp. are usually accompanied by mycotoxins contamination, which could cause health risks in humans. These fungal genera can produce several toxins such as FUMs (B1, B2), ZEN, trichotecenes (DON, nivalenol, T-2 toxins, H-T2), and other toxins [1].

## 3. Agro-Ecology, Climate, and Storage Conditions in Ethiopia

Ethiopia is situated in East Africa, located between 3°24′ and 14°53′ N and 32°42’ and 48°12′ E. The rift valley divides the country into two parts, forming the western and eastern highland plateau. The country has several mountains, hills, plateau, plains, valleys, and gorges, and the topography and elevation vary from the lowest point at Dankali depression (126 m below sea level) to the highest point at RasDashen Mountain (4620 m above sea level) [40]. Based on temperature, moisture, and elevation conditions, Ethiopia has 18 main agro-ecological zones [41], and owing to the country’s wide and complex topography, the environmental and climate conditions vary from one region to the other. These diverse agro-ecologies of Ethiopia allow production of a variety of crops as well as provide favorable environments for the growth of diversified mycotoxigenic fungi.

Cereals are the major food crops in Ethiopia. For instance, in the 2017–2018 growing season, maize (27.4%), teff (*Eragrostistef*, a small grain cereal endemic to Ethiopia) (17.3%), sorghum (16.9%), wheat (15.2%), barley (6.7%), finger millet (3.4%), rice (0.5%), and oats (0.2%) contributed to 87.48% of the country’s grain crop production [42]. Similar to other sub-Saharan African countries, crop contaminations both with mycotoxigenic fungi and their toxins are a severe threat to Ethiopia’s food security and food safety.

Ethiopian farmers, especially smallholding farmers, use traditional storage facilities, which are constructed using local materials. A significant number of farmers use underground-pit storage constructed inside the house or outside with wooden protective shade [43]. The pit dug is made to be narrow on the top, with the inside part covered with a slurry of cattle dung to prevent any leaks. Once the food grains are stored, the top part is sealed for several years until the products are required for consumption. In this kind of storage, although the risks of insect infections are low, stuffy and humid conditions as well as crop–soil direct contact favor the growth of molds and mycotoxin contaminations, resulting in inevitable food losses. Consequently, massive amounts (5–26%) of post-harvest food losses have been reported in Ethiopia, owing to poor food crop storage and a weak post-harvest system [44].

## 4. Occurrence of Mycotoxigenic Fungi and Their Toxins in Ethiopia

In Ethiopia, several researchers have reported the occurrences of mycotoxigenic fungi and their toxins in major cereals and other agricultural products (Table 1). In particular, special focus had been given to the occurrence of toxins that exceed the maximum limits set by EU countries, which are Ethiopia’s main trade partners. In a recent survey of 90 maize grain samples collected from West Showa and East Wallega zones, *Aspergillus* spp. were the most dominant fungi (50.7%) detected, followed by *Fusarium* spp. (26.4%), *Penicillium* spp. (22.3%), and *Trichoderma* spp. (1.07%) [45]. In addition, AFB1 was also detected in the range of 3.9–381.6 µg/kg. A total of 7.7% of maize samples had AFB1 content higher than the maximum limit (5 µg/kg) set by EU for maize to be use as an ingredient in foodstuffs [46], and higher prevalence of *Aspergillus* (94%), *Fusarium* (76.5%)*,* and *Penicillium* (64%) was detected in maize samples collected from smallholding farmers in different geographical locations [23]. Moreover, AFs (88%), FUMs (B1+B2) (2%), DON (29.4%), and NIV (17.7%) were detected in 17 samples, with 5.8% of the samples containing AFs (B1 + B2 + G1 + G2) that exceeded the EU maximum limit (10 µg/kg) for maize to be use as an ingredient in foodstuffs [46].

Another study conducted with 90 sorghum samples from eastern Ethiopia revealed that *Aspergillus* spp. and *Fusarium* spp. were the main cause of AFB1 and FUMs contamination, and reported higher load of *Aspergillus* spp. (1–2.5 log cfu/g) and *Fusarium* spp. (0.5–1.3 log cfu/g) [47]. Besides, AFB1 and FUMs were detected in 94% (0–33.1 µg/kg) and 71.1% (907–2041 µg/kg) of samples, respectively, and 2.22% of FUMs-positive samples were found to have higher levels of FUMs beyond the EU maximum limit (1000 µg/kg) set for other cereals like maize for direct human consumption [48]. A similar study revealed the occurrences of *A. flavus* (56.7%), *A. niger* (16.7%), and *A. parasiticus* (23.3%) in 30 sorghum samples collected from northern Shewa [49], and 96.6% of the samples were found to be contaminated with AFs.

Higher incidences of *Aspergillus* spp. (70–100%) have been reported in major peanut producing areas in eastern Ethiopia, and *A. flavus*, *A. parasiticus*, *A. caelatus*, *A. niger, A. tamarii*, and *A. ochraceus* were noted as co-occurring fungal species [25]. Besides, 33.8% of the total samples (160 peanut and 50 peanut cakes) were found to be positive for AFs. Likewise, other related studies have reported higher incidences of different mycotoxigenic fungal genera such as *Aspergillus* (*A. flavus*, *A. parasiticus*, *A. ochraceus*, *A. tamarii*,and *A. niger)* and *Penicillium* in groundnut collected from eastern Ethiopia [50,51]. For instance, Chala et al. [24] demonstrated that 93% of peanut samples from eastern Ethiopia had total AFs level (15–11, 900 µg/kg) beyond the EU maximum limit (4 µg/kg) for groundnut for direct human consumption [46]. A recent study by Worku et al. [52] reported the presence of total AFs (2.5–16.7 µg/kg) in 60% of wheat samples (n = 179) collected from different locations in Amhara, Benishangul-gumuz, Oromia, and Tigray regions. Among these samples, 50.8% had total AFs level beyond the EU maximum limit (4 µg/kg) for cereals intended for direct human consumption [46]. In addition, 17 samples were found to be positive for DON (350–1140 µg/kg), and about 3.4% of these samples presented DON content beyond the EU maximum limit (750 µg/kg) for cereals intended for direct human consumption [48].

**Table 1 toxins-12-00648-t001:** Summary of mycotoxigenic fungi and/or their toxins reported by some authors in Ethiopia.

Commodity	Location	Year of Study	Number of Sample (n)	Major Fungi Identified (Incidence; % or CFU/g)	Mycotoxin	Range or Mean (µg/kg or μg/l)	Positive Samples (%)	Test	Detections Limit (µg/kg) or LOD/LOQ	Incidence beyond EU Maximum Limit (%), (µg/kg)	Authors
Maize	Dire Dawa, Adama, Ambo	2004/2005	17	*Aspergillus* (94%), *Fusarium* (76.5%), *Penicillium* (64%)	Total AFs	<5–27	88	ELISA	1.75	5.8 (10)	[23]
Groundnut	Babile, Darolabu, Gursum	2013	120	-	Total AFs	15–11,900	93	ELISA	-	93 (4)	[24]
Maize	SNNP Region	2016	150	*Aspergillus* (75%), *Fusarium* (11%), *Penicillium* (8%), *Trichoderma* (6%)	Total AFs	20–91.4	100	TLC	-	100 (10)	[53]
Locally brewed beers	Addis Ababa	2015/2016	18	-	Total AFs	1.23–12.47	92	HPLC	-	16.7 (4)	[54]
Sorghum	Babile, Kersa. Haramaya	2013	45	*Aspergillus* (1–2.5 log cfu/g) *Fusarium* (0.5–1.3 log cfu/g)	AFB1	ND-33.1	94	ELISA	0.01–0.03	13.3 (10)	[47]
					Total FUMs	907–2041	71.1		0.01–0.03	2.22 (1000)	
Maize	Major growing areas	2011/2012	200	-	Total FUMs	25–4500	77	ELISA	0.025	7 (1000)	[55]
Sorghum	North Showa Zone of Amhara region state	2016	30	*A.flavus* (56.7%), *A.niger* (16.7%), *A.parasiticus* (23.3%)	Total AFs	11.44–344.26		HPLC	-	100 (4)	[49]
					AFB1	3.95–153.72	96.66				
					AFB2	1.17–91.82	93.33				
					AFG1	9.87–139.64	96.7				
					AFG2	3.22–52.02	90				
Wheat	Wenberma, Merawi, Ofla, Hetosa, Gedeb, Lemo.	2016	179	-	Total AFs	2.5–16.7	60.	LFIA	2	50.8 (4)	[52]
					Total FUM	330–710	16.2		250	-	
					OTA	2.1–148.8	20.1		2	4.5 (4)	
Maize	West showa and east wallega zones	2019	90	*Aspergillus* (50.7%), *Fussarium*(26.4%), *Penicillium*(22.3%) *Trichoderma*(1.07%)	AFB1	3.9–381.6	34.4	ELISA	-	7.7 (5)	[45]
Pre-milling ingredients	Amhara, Tigray, Oromia, SNNP	2018	126	-	Total AFs	4.57–3.50	64.3	ELISA	-	>2.3% (4)	[56]
Complementary foods						3.7–7.1					
Maize	South and southwestern Ethiopia	2015	100	*Penicillium* (80%), *Aspergillus* (75%) *Fusarium* (60%)	AFB1	9.3	8	LC-MS/MS	0.3		[33]
					AFB2	34	3		0.4		
					AFG1	64	6		0.3		
					AFG2	21	2		0.4		
					AFM1	18	2		0.3		
					FB1	606	70		3.2	-	
					FB2	202	62		2.4	-	
					FB3	136	51		2.4	-	
					FB4	85	60		2.4	-	
					FA1	37	34		2.4	-	
					FA2	32	35		2.4	-	
					ZEN	92	96		0.12	13.5 (100)	
					DON	221	42		1.2	-	
					NIV	91	45		1.2	-	
Teff flour	Addis Ababa city	2017	60	-	OTA	2	20	HPLC	0.78	3.3 (3)	[57]
Wheat flour						7.3	50		0.58	26.7 (3)	
Coffee	Jima zone	2014–2016	75	*Aspergillus* (79%), *Fusarium* (8%), *Penicillium* (5%)	OTA	0.03–22.9	64	ELISA	1.9/2 (µg/L)	5.5 (5)	[58]
Groundnut	Babile, Darolabu, Fedis, Gursum	2013–2015	160	*Aspergillus* (70–100%) *A. flavus*, *A. parasiticus*, *A. caelatus*, *A. niger, A. tamari*, *A. ochraceus*	AFB1	ND-2526	32	UPLC	1 (B_1_, G_1_) 0.05 (B_2_, G_2_)	-	[25]
					AFB2	ND-237					
					AFG1	ND-736					
					AFG2	ND-171					
Maize	West Gojjam	2015	30	*Aspergillus* (53.3–80%), *A. flavus, A. parasiticus,* *A. niger*	AFB1	7.43	78.7	HPLC	0.03	50 (4)	[50]
					AFB2	4.19			0.3		
					AFG1	14.1			0.014		
					AFG2	6.17			0.15		
					Total AFs	30.9					

Definitions: CFU: Colony forming unit; ND: Note detected; LOD: Limit of detection; LOQ: Limit of quantification; TLC: thin layer chromatography; HPLC: High performance liquid chromatography; UPLC: Ultra-performance liquid chromatography; ELISA: enzyme-linked immunosorbent assay; LC-MS/MS: Liquid chromatography with tandem mass spectrometry; LFIA: Lateral flow immunoassay.

Likewise, another study performed with 150 maize samples collected from Southern Nations Nationalities and Peoples (SNNP) regions also reported the occurrences of *Aspergillus* spp. (75%), *Fusarium* spp. (11%), *Penicillium* spp. (8%), and *Trichoderma* spp. (6%) in the samples [53]. In addition, higher incidence of AFs-producing fungi, evidenced by detection of AFs in the range of 20–91.4 µg/kg in 100 samples, implied the presence of a high level of AFs in the samples beyond the EU maximum limit for maize to be use as an ingredient in foodstuffs [46]. Multi-mycotoxins and fungal analyses performed on 100 maize samples from south and southwestern Ethiopia [33] revealed that all maize samples (100%) were contaminated with various types of mycotoxigenic fungi. In particular, *Penicillium* spp. were observed to be dominant (80%), followed *Aspergillus* spp. (75%) and *Fusarium* spp. (60%). Besides, AFB1, B2, G1, G2, and M1 were detected in 2–8% of the total samples, whereas FUMs B1, B2, B3, B4 were found in 70%, 62%, 51%, and 60% of the samples, respectively. Although FUMB1 was the major contaminant, with a maximum concentration of up to 11,831 µg/kg, ZEN was the most common mycotoxin (96%), with 13.5% of the samples presenting high levels of ZEN beyond the EU maximum limit (100 µg/kg) for maize intended for direct human consumption [48]. 

A similar multi-mycotoxin analysis was conducted by Chala et al. [59], who identified a higher number of metabolites (84), including *Aspergillus* metabolites (23%), *Fusarium* toxins (20%), and *Penicillium* metabolites (13%), along with the main mycotoxins (19%), in sorghum samples collected from South, East, and Northwest Ethiopia. Moreover, considerable amounts of AFB1, B2, G1, and G2 were also detected, and the mean levels of AFB1 (29.5 µg/kg) and G1 (29.7 µg/kg) detected in sorghum samples were beyond the corresponding EU maximum limits for cereals for direct human consumption [46]. Despite its negligible amount, AFM1 was also detected (0.89 µg/kg) in 1.43% of the total sorghum samples (n = 70). AFM1 is hydroxylated form of AFB1, it is not common to find it in foods other than dairy products. However, it has been reported that trace amounts of AFM1 in *A. flavus* and *A. parasiticus* contaminated crops could be detected when there is higher level of AFB1 contamination [60]. Studies in other African countries reported the detection of AFM1 in maize [61] and cassava [62]. Furthermore, higher incidences of ZEN contamination were observed on sorghum (32.9%) and finger millet (51.5%), when compared with other common mycotoxins contamination.

AFs contamination has also been reported in spices and their mixes that are common ingredients in Ethiopians daily foods. Detection of AFB1 in Shiro is of significant concern because Shiro is consumed by all Ethiopians. Shiro is prepared by mixing legumes and several spices, including chili pepper, ginger, fenugreek, cumin, coriander, cloves, cardamom, nutmeg, and others. However, despite roasting the legumes, the temperature used for roasting cannot destroy the toxins. A high content of AF B1 (100–500 µg/kg) was reported in 8.3% of the studied samples [63], indicating that household heat treatment does not ensure AFs removal if the raw materials are contaminated.

In addition to AFs, higher OTA contamination has been detected in both Ethiopian daily staple foods and daily consumable beverages. Ayalew et al. [64] found that 24% and 27.3% of cereals and teff samples were positive for OTA (2.7–2106 µg/kg), respectively. Another study by Geremew et al. [57] reported the presence of OTA in 20% of teff flour collected from Addis Ababa. These results indicate that OTA contamination is a serious health concern for Ethiopia because *injera* (flat bread made out of teff) is a daily meal for majority of people in Ethiopia. Besides, 50% of wheat flour samples were noted to be contaminated with OTA, of which 26.7% of the samples had OTA contents higher than the EU maximum limit (3 µg/kg) for cereals intended for direct human consumption [46]. It must be noted that OTA is considered to be nephrotoxic and immunosuppressive [65], and has been classified as a compound with possible carcinogenicity to humans by the International Agency for Research on Cancer (IARC) (Group 2B) [66].

Mycotoxigenic fungal contamination of several export commodities, including high-value cash crops such as coffee, which is one of Ethiopia’s primary export commodities, is becoming a serious problem. In a study to determine mycotoxigenic fungal contamination of coffee samples collected from potential coffee-producing areas, *Aspergillus* (79%), *Fusarium* (8%), and *Penicillium* (5%) were noted to be the predominant toxigenic genera [58]. Similarly, Biru and Tassew [67] also reported the occurrences of ochatoxigenic species, such as *Aspergillus ochraceus* and *Aspergillus westerdijkiae,* in *Coffee arabica* samples collected from Jima zone of Oromia region in southwestern Ethiopia. Moreover, a study by Geremew et al. [58] revealed the presence of OTA in the range of 0.03–22.9 μg/kgin 64% of the coffee samples collected from the most important coffee-producing zones. Furthermore, 5.5% of the samples from wet and dry coffee processing were noted to be contaminated with OTA, which was beyond the EU maximum limit (5 μg/kg) for coffee for direct human consumption [46]. The occurrence of OTA on coffee is of main concern in Ethiopia owing to its significant impact on both economy and human health. Coffee is the highest economically valued cash crop of Ethiopia accounting for 26% of the country’s export earnings [68], with close to 15 million Ethiopians being dependent on coffee sector [69]. Besides, Ethiopia is not only the origin and Africa’s top producer of *C. arabica,* but also the biggest consumer. In terms of per capita consumption, Ethiopia is the largest coffee-drinking country in Africa and the world. About 50% of coffee produced in Ethiopia is consumed locally [70]. However, most of the coffee treated and consumed locally in Ethiopia is lower grade coffee that has been rejected for export or locally processed with low standard to meet local demand. Thus, the local consumers could be exposed to higher OTA level, because coffee beans with defects have a higher possibility of *Aspergillus* and associated OTA contamination [67]. 

In addition, mycotoxins have also been reported in industrially processed alcoholic beverages in Ethiopia. For instance, in a survey of 18 local beer brands for AFs contaminations, 92% of the samples were found to be positive for AFs with a mean AFs value of 3.52 µg/L (range: 1.23–12.47 µg/L) [54]. Of these AFs-positive samples, two (16.7%) beer brands were found to have AFs contents beyond the national permissible limit for AFs (10 µg/kg) [26]. Similar to other cereals, barley, which is the main ingredient for beer production in Ethiopia, is also susceptible to mold attack and mycotoxins contamination. In most of the reported studies, co-occurrence of *Aspergillus* and *Fusarium* toxins, including AFs, FUMs, ZEN, DON, NIV, and other metabolites, was commonly noted in major Ethiopian cereal crops (Table 1). In particular, co-occurrence of mycotoxins is a major concern because the possible synergetic toxicities are not well known [71].

## 5. Public Health Impacts of Mycotoxins in Ethiopia

Mycotoxins could cause acute or chronic health impacts on humans. Among the major mycotoxins, the health impacts of AFs have been well documented in Africa. For instance, from 2004 to 2006, several hundred Kenyan died from acute AFs poisoning [72,73]. Some chronic health complications resulting from AFs exposure include stunting, immunosuppressive effects, and cancer [74,75]. Since 2002, AFs have been recognized as Group 1 human carcinogens by the IARC [66], and can act synergistic with HBV to increase the possibility of chronic liver diseases (CLD) or hepatocellular carcinoma (HCC) [15,76,77]. A quantitative AFs-related HCC risk assessment performed by Liu et al. [15] based on nation’s food consumption patterns, AFs food prevalence, HBV prevalence, and population size revealed that AFs exposure can be attributed to 25,200–155,000 (4.8–28.2%) HCC cases annually around the world, with most of the cases occurring in the sub-Saharan African region where there is a higher incidence of HBV. 

In Ethiopia, about 11–288 HCC cases were attributed to chronic aflatoxin exposure, and 21–643 HCC cases were attributed to synergetic effects of AFs and HBV, with chronic HBV prevalence in Ethiopia being the highest among African countries (6–7%) [78]. Furthermore, Ayalew et al. [64] estimated AFs exposure (1.4–36 ng/kg body weight/day) based on AFs level (26 µg/kg) in staple cereals. In a cross-sectional study conducted in eastern Ethiopia, chronic HBV infection was reported in 33% of patients with CLD; however, in more than 50% of the patients, the etiology of the liver disease was unexplained [79]. Hence, it is necessary to perform epidemiological studies in an area with higher prevalence of AFs contamination to determine the possible public health impacts that can be attributed to AFs exposure [24,25,47]. Moreover, in Ethiopia, the mortality rate owing to HCC is among the highest in the world, reaching 93.4 per 100,000 [80]. Therefore, it is necessary to ascertain the possible correlation between AFs exposure and CLD in Ethiopia and develop appropriate mitigation strategies to control AFs contamination. 

Similar to *Aspergillus* mycotoxins, the prevalence of *Fusarium* mycotoxins has also been noted to be higher in Ethiopian commodities, causing higher health risks owing to dietary *Fusarium* mycotoxins exposure. Previous studies [33,47,55,59] have confirmed higher human exposure to *Fusarium* toxins in Ethiopia. Elevated *Fusarium* toxins exposure is known to cause serious health hazards both for humans and animals. According to the IARC, FB1 is possibly carcinogenic to humans (Group 2B) [66], and many studies have reported a correlation between dietary FUMs exposure and esophageal cancer (EC) [81,82]. A survey conducted in 2013–2016 on the incidences of EC across Oromia region in Ethiopia showed that two-thirds (68%) of EC cases (among 669 total EC cases in this region) originated from Arsi and Bale zones of Oromia Regional State, which are the main cereal producers in this region [83] Similarly, another study also concluded that consumption of hot cereal-based porridge had a positive link to higher EC prevalence in the area [84]. Hence, further epidemiological studies are required to determine the associations among consumption of hot cereal-based diet, FUMs exposure, and EC cases.

## 6. Early Age Mycotoxins Exposure and Malnutrition in Ethiopia

Children are vulnerable to dietary mycotoxin exposure at early childhood. For instance, the AFs exposure started at the infant stage from breast milk if the mother is under dietary exposure [85]. Weaning foods (milk, complementary foods) made from contaminated products are also sources of exposure [56]. A survey of AFs contamination performed using 126 food samples (complementary foods and ingredients) collected from Amhara, Tigray, Oromia, and SNNP regions revealed that 121 samples (96%) were positive for total AFs (0.5–12.4 µg/kg), with about 80% of the samples containing total AFs exceeding the EU maximum limit (4 µg/kg) set for cereals intended for direct human consumption [46]. As children are given weaning and adult food at early age, mycotoxins exposure beings at early age, and common cereals such as maize, sorghum, and millets, and other foods such as peanuts are highly susceptible to mycotoxins contamination unless properly dried, harvested, stored, and processed. 

Mycotoxins (e.g., AFs) exposure could exacerbate malnutrition, and a potential correlation between AFs exposure and childhood stunting has been reported [86]. Moreover, recently reduced rate of gestational weight gain was attributed to AFs exposure [87].Similar to HCC, childhood stunting is most prevalent in South and East Asia and sub-Saharan Africa, where foodborne AFs exposure is high [88]. However, in Ethiopia, the synergy between mycotoxins exposure and malnutrition has not been well documented. Only one epidemiological study on AFs exposure among children (1–4 years of age) in Amhara and Tigray regions had been published, although the study findings did not show any associations between malnutrition in children and AFs exposure. Nevertheless, the presence of AFs (B2, G2, M1) in 17% of the urine samples from children indicated that AFs exposure is a severe childhood health concern in Ethiopia [89]. Moreover, in Ethiopia, childhood malnutrition is higher, with 38% of children (<5 years of age) being stunted [90]. Therefore, there is an urgent need for more epidemiological studies to better understand the synergy between mycotoxins exposure and malnutrition as well as to implement regulatory measures to control AFs contamination in infant foods and maternal nutrition.

## 7. Mycotoxins and Their Economic Impacts in Ethiopia

In Ethiopia, the scale of the economic impact of mycotoxins contamination has not been well documented. Nevertheless, existing reports show the impacts of mycotoxins contamination on export market of Ethiopia. In 2015 (February–December), the EU Rapid Alert System for Food and Feed (RASFF) issued four alerts for OTA levels (92.5–139 μg/kg) in different spices products from Ethiopia and rejected the products thrice at the borders of Germany and the UK [35]. Similarly, according to RASFF reports, in 2017–2019, 18 border rejections were issued on spice products from Ethiopia owing to higher levels of mycotoxins (AFs or OTA) and lack of certified analytical reports [91]. These products were destined to different EU countries, including France, Italy, Luxemburg, Sweden, and Norway. However, the refusal of Ethiopian agricultural products by the EU had been inadequately covered by the local media in Ethiopia. The newspaper, Capital Ethiopia (16 July 2018), reported that hot pepper powder worth USD 10 million had been returned to Ethiopia from the EU markets owing to the presence of unsafe levels of AFs and OTA [92]. On 21 December 2016, the EU imposed special conditions urging Ethiopian spice exporters to provide additional guarantees [93], which discouraged spice exporters. As a result, the export volume of goods from Ethiopia to the EU has dramatically reduced with exporters shifting their commercial activities to other countries with less strict mycotoxins standards, but relatively cheaper market [94]. Consequently, Ethiopia has been losing its EU market, with considerable revenue drop in the past few years. For example, according to the Ethiopian Reporter newspaper (12 August 2017), the Ethiopian revenue from spice export declined by 5% in 2017, when compared with that in the preceding year [95].

In addition, mycotoxins contamination also affects local food processing industries that obtain raw material supplies from local producers and smallholding farmers. As one of the challenges of commercialization of local products is the presence of unsafe levels of mycotoxins, local Ethiopian industries have begun to import quality agricultural raw materials. However, import of raw materials could also have an effect on the escalating burden of foreign currency, which is the country’s critical problem [96]. Overall, AFs contamination can cause direct and indirect economic impacts on Ethiopia, as well as affect the livelihood of smallholding farmers by reducing the quality of products, which could result in price discounts and export ban. 

## 8. Mycotoxins and Their Impacts on Livestock Production in Ethiopia

Ethiopia has the largest livestock resources in Africa [97]. In 2011–2012, the country had a total cattle population of 52 million, along with 24.2 million sheep, 22.6 million goats, and 45 million chickens [98]. The livestock sector contributes to 12–16% and 30–35% of the total and agricultural GDP, respectively [68]. The quality of feed is one of the determinants of livestock health and productivity, and cereal grains, primarily corn, are widely used in animal feed. In Ethiopia, oilseed cake (e.g., noug and linseed cake) is mainly utilized as feed, which is highly susceptible to AFs contamination [99]. Owing to lack of awareness and/or economic reasons, animal husbandry smallholders do not give adequate importance to feed safety. As a result, household leftover foods and highly moldy cereals that are not suitable for human consumption are usually used as animal feed [17]. In addition, household herders generally feed their animals spent grains (*atela*) from household alcoholic beverages (*arake* or *tela*) that are prepared from moldy grains [94].

Although there are no published reports on animal health risks associated with feed contamination in Ethiopia, studies around the world have shown adverse health hazards of feed contamination on animals such as swine, cattle, and poultry [6]. Moreover, the presence of transformed mycotoxin metabolites in livestock products (e.g., milk, meat, egg) is of serious food safety concern. Nevertheless, reports on mycotoxins contamination on livestock products in Ethiopia are limited, with only two reports published about the prevalence of mycotoxins in feed and milk. A study by Gizachew et al. [99] revealed that 93% of milk samples collected from Addis Ababa milk producers were contaminated with AFM1, beyond Ethiopian and EU maximum limit (0.05 µg/L) [46], Table 2. The same study reported that all the examined feed samples were contaminated with AFB1 (7–419 μg/kg), with 88.8% of feed samples containing AFB1 exceeding the maximum limit for AFB1 (5 μg/kg) set by the EU [100]. When these findings were publicized by the local media in Ethiopia, it created fear among families around Addis Ababa, and the milk value chain was seriously affected for a short period of time until officials and media offered some awareness to the society. Moreover, in December 2017, a magazine named Addis Standards reported that the Ethiopian Food and Drug Authority (FDA) ordered disposal of 129,000 L of milk (average of 3000 L/day) produced in one of the milk processing farm in Ethiopia owing to AFM1 contamination [101]. Although this incident had a significant implication on the performance of the company as well as livelihood of nearby smallholding dairy farmers who supplied milk to the company, it revealed that the milk production value chain of Ethiopia is highly vulnerable to AFs contamination.

## 9. Mycotoxins Regulation in Ethiopia

In Ethiopia, the Food and Drug Regulation and Control System has been enacted by Drug Administration and Control Proclamation No. 176/1999 and Council of Minister Regulation No. 299/2013, with main responsibility given to Food Medicine and Healthcare Administration and Control Authority (Ethiopian FDA), and some of the responsibility shared among sector ministries such as Ministries of Health, Agriculture, Trade, Industry, Innovation, and Technology. In 2019, the Ethiopian FDA implemented the Food Registration Directive, whereby issues related to food safety and contamination as well as appropriate measures based on the legislation have been clearly indicated. Moreover, several agencies such as the standard agency and commodity exchange agency accountable to various ministries are also directly or indirectly involved in Ethiopian food legislation and regulation tasks. As a result, mandate overlapping and lack of coordination with regard to mycotoxins issues across these agencies are inevitable. Besides, most of the agencies do not consider mycotoxins level as a quality parameter or criterion for inspection and certification of local food establishments.

Globally, many countries have stringent mycotoxin standards to reduce healthcare costs and access high-value international markets. According to FAO’s worldwide survey in 2003, over 100 nations have mycotoxins regulations for food and feedstuffs [102]. Table 2 shows the maximum limit for mycotoxins with the corresponding food items stated by the EU countries. The development of mycotoxins regulations is a long process and capital-intensive, with several factors or driving forces affecting this process, such as availability of toxicological data, exposure data, analytical methods, and knowledge about the distribution of mycotoxins and legislation in other trade partner countries [103]. Currently, the Ethiopian export market to the EU is facing several challenges concerning detection of excessive levels of mycotoxins in products. Hence, the state must immediately start to work on capacity developments to generate mycotoxins exposure data to establish standard mycotoxins limits. Until 2003, Ethiopia did not have any regulation for mycotoxins levels in any food products [102]. According to the Ethiopian Standard Agency (ESA) and Ayelign et al. [26], very few commodities are currently regulated for mycotoxins contents (Table 2). 

## 10. Analysis of Mycotoxins in Ethiopia

Dietary exposure to mycotoxins, even at low levels, has been confirmed to be very dangerous for health. Therefore, the demand for sensitive analysis and quantification methods is higher [104]. Moreover, the international market requires licensed analytical reports and health certificates for export products. Hence, domestic precedence of accredited and qualified laboratory facilities is important. Several mycotoxins detection methods are currently available, with modern and efficient techniques being continuously developed. High-performance liquid chromatography (HPLC) is known for its sensitivity and accuracy for detecting several mycotoxins. In recent years, a more sensitive and fast method called ultra-performance liquid chromatography is being used [105]. Currently, chromatographic methods combined with mass spectrometry (LC/MS) are being employed to detect and quantify mycotoxins [104]. Besides commercially available ELISA with narrow detection range, lateral flow immunoassay (LFIA) kits also provide a relatively easy assay for quantifying mycotoxins [106]. In Ethiopia, several methods, including TLC, ELISA, LFIA, HPLC, LC/MS, LC-MS-MS, and UPLC are employed for mycotoxins detection and quantification. However, most of the modern methods (HPLC, LC/MS, LC-MS-MS, UPLC, etc.) reviewed in this study had not been conducted in Ethiopia, but were performed in collaboration with other international institutions. Thus, as a part of the food control strategy, accredited and qualified laboratories are needed in Ethiopia to improve the export market, which is the nation’s key economic backbone. 

## 11. Raising Awareness and Adulteration Practices in Ethiopia

In Ethiopia, public understanding about molds and mycotoxins is generally limited. A study evaluating the knowledge on mycotoxins and their health impacts revealed that many of the Ethiopians are not even aware of the effect of consuming moldy products [107], and that farmers and traders have no significant knowledge about the health risks related to long-term AFs exposures. 

In general, majority of Ethiopians consider molds on food as harmless. Besides, it is believed that molds only affect the organoleptic properties of food, and can be easily treated by household cooking, drying, or fermentation. Beyene et al. [17] performed a survey to identify Ethiopian mothers’ knowledge and practice with respect to AFs contamination in complementary foods, and found that almost one-fourth of the respondents use moldy cereals to make local beverages such as *tela* (locally made beer) and *arake* (locally made spirit). Similarly, moldy *injera* (fermented flat bread mainly made of Teff) is usually dried under sun or heated to form *injera* crumb (*dirkosh*) [94]. These traditional practices of food and beverage production could worsen AFs contamination among local people. 

Furthermore, intentional adulteration and poor storage practices are commonly followed by fraudulent traders in Ethiopia [108]. For instance, addition of water to dry commodities such as peanut and spices to increase weight is the most frequently employed malpractice among local traders, which is hazardous because the conditions of firmly packed adulterated product is suitable for mold germination, growth, and mycotoxins production [94,108]. Recently, the Ethiopian Biotechnology Institute (EBTi) had developed a training manual that could help to create awareness along crop value chain, and includes pre- and post-harvest practices with a potential implication for fungal development that causes mycotoxins development in crops, as well as presents mitigation strategies [109].

## 12. Mycotoxin Management Strategies

While there is no single and best mycotoxin intervention strategy to address mycotoxin contamination, multifaceted approaches at different stages in the value chains are encouraged [110]. Pre- and post-harvest mitigation strategies, including good farming practice (GFP), good agricultural practice (GAP), and good handling practice (GHP) [75], mainly used to prevent mycotoxins contamination are the most promising because they are cost-effective and easy to use by farmers. Besides, proper fertilization [111], selection of disease-resistant breeding [112], cropping rotation systems [113], use of fungicides/pesticides, insects control measures, proper drying, appropriate storage, and transportation are encouraged. In addition, physical sorting methods, performed by removing damaged products based on quality parameters such as color, size, shape, density, and visible fungal growth [110], are also known to have the potential to reduce the impacts of both mold infection and mycotoxins contamination. Moreover, the use of biocontrol agents has emerged as a promising approach. For example, the most successful biological control method to reduce AFs contamination is the application of non-aflatoxigenic *A. flavus* strains to soils, which competitively exclude naturally occurring toxin-producing *A. flavus* strains in the field, resulting in remarkable decrease in AFs production [114,115]. 

Several countries in West Africa (Senegal, Gambia, Burkina Faso, Ghana, and Nigeria), East Africa (Kenya, Uganda, and Tanzania), and southern Africa (Mozambique, Malawi, and Zambia) are in the process of developing and using native *aflasafe* under the umbrella of International Institute for Tropical Agriculture (IITA) [116]. In Ethiopia, two initiatives are in progress. The newspaper, Reporters (posted on 25 August 2018), revealed that EBTi has launched a project with the objective to develop biocontrol agents from native *A. flavus* strains isolated from homegrown crops [117]. Capital newspaper (posted on 16 July 2018) reported that Ethiopia has also planned to use *aflasafe* for red peppers in combination with Ministry of Agriculture and IITA [92]. However, no biocontrol products are currently being used in Ethiopia. 

Ethiopia is known for its largest livestock population, and the quality of different livestock products (dairy products, meat, eggs) in terms of mycotoxins contents must be maintained. Mycotoxin binders mainly used as feed additives could be a reliable option for mycotoxins mitigation. Adsorbents that are usually added to compound animal feed have been confirmed to reduce mycotoxins [118,119], and binders such as naturally occurring clays (e.g., bentonite) could sequester mycotoxins in the feed or intestinal tract after being consumed by animals. Consequently, the harmful effects of mycotoxins on the animals are reduced [120] and the transfer of mycotoxins to animal products such as milk, meat, and eggs is decreased [121,122]. Moreover, several in vitro studies have also reported 35–93% decrease in AFB1/AFM1 level by the action of bentonites [123,124]. Currently, bentonite-containing feed additives are being widely used in the diets of lactating dairy cows, and have been proven to be safe for use as animal feed additives by the USA Food and Drug Administration [125].

According to the report by Ethiopian Geological Survey Institute, Ethiopia has huge bentonite resources available in different locations of Afar and Oromia regions [126]. Moreover, our informal communications with the locals revealed that the slurry of bentonite is traditionally consumed to treat stomach discomfort. The EBTi had investigated the application of bentonite and zeolite to minimize AFs contamination in milk, and found a significant reduction in AFs level; however, it was noted that bentonite and zeolite must be further processed for commercial applications [109]. Hence, more studies are recommended to better understand the efficacies and safety of bentonite for future use as a mycotoxin mitigation alternative.

## 13. Conclusions and Recommendations

Mycotoxigenic fungi and their toxins are prevalent in major cereals (teff, wheat, maize, sorghum, and barley) from potential agricultural regions (Oromia, Amhara, and SNNP) of Ethiopia. These cereals are the core of Ethiopia’s agriculture and food economy, accounting for 14% of the total GDP (2005/06) and 64% of the calories consumed by Ethiopians [127]. Therefore, occurrences of mycotoxigenic fungi and their toxins could be considered as a serious problem both in crop production and animal husbandry. Moreover, unsafe levels of mycotoxins significantly affect the export market of Ethiopia, and export trade is challenged by several bottlenecks, including lack of accredited laboratory for mycotoxins tests, delayed authorization procedures, etc., resulting in the loss of millions of dollars. Although studies on the actual human and animal health implications of mycotoxins exposure are limited, the prevalence data generated so far are adequate to demonstrate the existing public health risks in Ethiopia. Therefore, to alleviate these problems, use of pre- and post-harvest mitigation strategies (GFP, GAP, and GHP) that prevent mycotoxins contamination of value chains is recommend. Besides, research on AFs biocontrol development and mycotoxin binding technologies should also be encouraged because these technologies are affordable. Moreover, human resources and infrastructure development is necessary, and advanced facilities that accelerate epidemiological studies should be made available in public institutions. In addition, a nationwide in-depth mycotoxin risk assessment should be performed for a better understanding of the economic and health risks associated with mycotoxins contamination. Furthermore, higher priority must be given to establish more accredited public laboratories in Ethiopia, which will improve authorization and certification, thus accelerating export performance. Sustained public awareness such as training of value chain actors and consumers is necessary to develop a population that is well informed about fungi and mycotoxins as well as their economic and health risks. Public awareness activities via different tools such as media outlets, public gatherings, and social learning must be employed to create awareness about mycotoxins. Lastly, as Ethiopian women are highly involved in household food preparation and taking care of children who are highly vulnerable to mycotoxins exposure, gender roles that are important in making decisions on food handling at the household level should not be overlooked.

## Figures and Tables

**Table 2 toxins-12-00648-t002:** Maximum limits of mycotoxins (µg/kg) in Ethiopia.

Commodities	Aflatoxins			Fumonisins (FUMs)
	AFB1	Total AFs	AFM1	
Peanut kernels	5	10		-
Peanut butter	4	10		-
Maize	5	10		-
Wheat	-	4		-
Sorghum	5	10		-
Barley	5	10		-
Teff flour	-	4		-
*Injera*	5	10		2000
Raw cow milk	-	-	0.05	-
Lentil	5	10		-
Chickpea	5	10		2000
Whole peas	5	10		2000
Split peas	5	10		2000
Dry beans	5	10		2000
Dry faba beans	5	10		2000
Kolo (roasted grain, ready-to-eat food)	-	-		2000
Beer	5	10		-

Sources: [26] and ESA (personal communications).

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
