# Peer review of "Mycotoxins in Ethiopia: A Review on Prevalence, Economic and Health Impacts"

_toxins, 2020, doi:10.3390/toxins12100648_

Round 1

Reviewer 1 Report

English use should be thoroughly revised.

Some references are rather old (more than 10-15 years) for a review paper: i.e. 4, 6, 7, 9, 12 in just the first paragraph of Introduction. Could you look for more recent/up-to-date references?

Lines 59-60: Which EU-harmonized standard (April 2002) is being referred to?

Line 99: ...usually belong to...

Line 107: check the reference as most Aspergillus species can grow well below aW 0.98

Line 110-111: What do you mean by ‘rarely they (A. flavus and A. parasiticus) are also able to produce aflatoxin M1’? The production of aflatoxins B1, B2, G1 and G2, as well as M1 should be clarified by the authors.

Line 113: typo in nige (niger is correct)

Line 117: revise the aW range for Fusarium growth and toxin production

  1. Mycotoxigenic fungi and growth conditions: Authors only describe Aspergillus and Fusarium molds. What about Penicillium? (and others such as Alternaria, Claviceps, Stachybotrys, Trichoderma...)

Lines 139-141: What do the percentages in parentheses represent? (area or production-Tm)

Line 158: How come that Table 2 is mentioned first than Table 1?

Lines 159-168: EU establishes maximum levels for mycotoxins in foodstuffs (not recommended limits). Revise terminology throughout manuscript.

Lines 178-182: Revise legitimate fungal names and correct typos

Lines 217-223: What spices are commonly used in Ethiopian shiro?

Line 229: typo in westerdijikiae (correct is westerdijkiae). Revise fungal names throughout manuscript (use MycoBank database for correct names).

  1. Occurrence of mycotoxigenic Fungi and their toxins in Ethiopia. Authors mention several studies on mycotoxin contamination carried out in Ethiopia, but the analytical methodology used could be mentioned here (not in section 9).

Lines 239-241: A mention that aflatoxins are classified as Group 1 carcinogens by IARC should be added.

Lines 247-249: Do you mean HCC cases per year of what?

Lines 260-268: What cereal/s is/are used in Ethiopia to make beer?

Line 324: What are complimentary foods? Do you mean complementary foods for infants?

  1. Public health impacts of mycotoxins in Ethiopia. This section contains data on mycotoxin occurrence that belongs to section 4. The information is somewhat messy.

Line 338: Could you check the RASFF on mycotoxins from Ethiopia for a more recent year than 2015?

Line 362: Please provide a reference to support this.

Lines 433-434: Could you make a table (or a paragraph) with the maximum levels for mycotoxins in foods that are established by current Ethiopian regulations?

Table 2: Please revise COMMISSION REGULATION (EC) No 1881/2006 of 19 December 2006 setting maximum levels for certain contaminants in foodstuffs (and later amendments i.e. consolidated version) to update and complete the information on current maximum levels

Reviewer 2 Report

Summary

The authors present a review on mycotoxin in Ethiopia.  This focuses on the prevalence of both the toxins and the mycotoxigenic fungi.  Later sections discuss health impacts on the population and livestock, and also economic implications for trade and regulatory compliance.

General Comments

Overall, the major value of the paper is the careful work the authors have done to describe the current state of research on mycotoxins in Ethiopia specifically.  In most sections the articles reviewed do, in fact, give data specific to Ethiopia.  And when the authors review information from other sub-Saharan Africa studies, or EU/US studies, they do appropriately contextualize and contrast.  My three major comments for improvement are:

  • Revise the tables to reflect the strength of this work as being about Ethiopia.
    • As written, Table 1 presents which standard toxin analysis methods are used in Ethiopia. But the parameters described are basic methods parameters.  So it is not clear what value this table has for understanding the Ethiopian context.  For example, why would the ‘detection limits in Ethiopia’ be any different than anywhere else the methods are conducted.  Consider removal or revision.  For example, a useful table might be what testing capacities currently exist in specific labs in Ethiopia, and references to papers where those analyses are published.  That would allow an international audience to appreciate the world-class science that can be done in-county.
    • Similarly, as written Table 2 presents EU mycotoxin regulatory limits. There is no obvious reason to include that previously published data in this review.  Consider removal or revision. One suggestion would be to expand the table to explicitly contrast the regulations in Ethiopia.  This would support the reference in lines 433-434 and allow a reader to know the actual regulations in-country with the same specificity as what is presented for the EU.
  • Consider a summary table for the papers reviewed in section 4, lines 155-234. The authors, presumably, correctly identify a literature gap on mycotoxigenic fungi earlier in the introduction.  IF so, then a table reviewing papers that fill this gap would be very helpful.  It would also cut down on the text that is very dense and hard to read, and allow the text to reflect important generalizations such as in lines 231-244.
  • The paper has numerous grammatical errors in written English that do not affect the ability to understand the science but still present an unprofessional document. For example: lines 76-77 is: In Ethiopia, studies on mycotoxins have been done for more than four decades, with its earlier aflatoxin report date back in 1981 [25]. but should be:  In Ethiopia, studies on mycotoxins have been done for more than four decades, with [the] earlie[st] aflatoxin report date[d] [from] 1981 [25]. This reviewer’s understanding is that Toxins does not provide copyediting services.  So the authors should secure line-by-line copy editing by a reviewer fluent in technical English.  

Line Item Comments

266: Estimating toxin exposure based on consuming 6 standard beers a day seems like a lot of beer. Maybe use a more reasonable consumption level?

362: Probably need a reference to justify Ethiopia having the largest livestock resources in the continent.

468: One other category of management strategies is remediation if contamination does occur.  In this class, physical sorting probably has the most potential.  Consider adding.

495: ‘US’ Food and Drug ‘Administration’.

533: Consider removing the ‘Materials and Methods’ section.  As written, this is not a systematic review.  That would require much more description of the process to filter search results as well as systematic summary.  It is possible the table suggested in the major comments would move towards a systematic review, but that would likely be a larger revision than is desired by the authors.

Round 2

Reviewer 1 Report

Lines 58-61: Delete the sentence because it is not up-to-date and refers to regulations that are no longer in force. Commission Regulation No 466/2001 is no longer in force, the date of end of validity was 28/02/2007 and it was repealed by current Commission Regulation (EC) No 1881/2006. Delete as well References 16 and 17.

Lines 106-110: The misconception about water activity and mold growth still remains. The paragraph should be revised with care. Please revise cited reference 32, which by the way is misquoted. Correct reference is: Agriopoulou, S., Stamatelopoulou, E., & Varzakas, T. (2020). Advances in occurrence, importance, and mycotoxin control strategies: prevention and detoxification in foods. Foods, 9(2), 137.

Line 110-112: The misconception about the production of the different aflatoxin types still remains. Aflatoxin M1 is the hydroxylated metabolite of aflatoxin B1 and can be found in milk or milk products obtained from livestock that have ingested contaminated feed. Aflatoxin M1 is not directly produced by molds, but by animals fed contaminated diets.

For coherency, all references to maximum levels in the manuscript should be made for either unprocessed products or for products for direct human consumption, but not a mixture of both. Authors should obtain latest consolidated version of COMMISSION REGULATION (EC) No 1881/2006 for proper quotations.

Line 173: the maximum level of fumonisins in maize is not 2000 ppb

Line 230: the maximum level of zearalenone in maize intended for direct human consumption is not 350 ppb

Table 1: typo in complimentary foods (correct complementary)

Line 336: the maximum level of aflatoxin B1 in processed cereal-based foods and baby foods for infants and young children is 0.1 ppb (COMMISSION REGULATION (EC) No 1881/2006 of 19 December 2006 setting maximum levels for certain contaminants in foodstuffs)

Still uncorrected: EU establishes maximum levels for mycotoxins in foodstuffs (not recommended limits, permissible limits, etc). Revise terminology throughout manuscript.

Reference 49 is incomplete:

COMMISSION REGULATION (EC) No 1126/2007 of 28 September 2007 amending Regulation (EC) No 1881/2006 setting maximum levels for certain contaminants in foodstuffs as regards Fusarium toxins in maize and maize products

Author Response

Reviewer one comments, author’s responses and manuscript changes

Thank you! We found your comments extremely helpful and have revised accordingly. Our responses are given in a point -by-point manner below. Changes to the manuscript are shown by a blue highlight.

Comments and Suggestions for Authors

Comment 1: Lines 58-61: Delete the sentence because it is not up-to-date and refers to regulations that are no longer in force. Commission Regulation No 466/2001 is no longer in force, the date of end of validity was 28/02/2007 and it was repealed by current Commission Regulation (EC) No 1881/2006. Delete as well References 16 and 17.

Response: Thanks for your comment. The sentence has been deleted and references 16 and 17 removed.

Comment 2: Lines 106-110: The misconception about water activity and mold growth still remains. The paragraph should be revised with care. Please revise cited reference 31, which by the way is misquoted. Correct reference is: Agriopoulou, S., Stamatelopoulou, E., & Varzakas, T. (2020). Advances in occurrence, importance, and mycotoxin control strategies: prevention and detoxification in foods. Foods, 9(2), 137.

Response: Thanks for your generous comments about water activity, we have revised the concept with care by citing other references and the new paragraph highlighted with blue color is read as “Many species in the genera Penicillium and Aspergillus can be described as moderate xerophiles. Xerophilic fungi are characterized as being capable of growing under conditions of reduced aw (< 0.85), and are most commonly associated with dried foodstuffs like cereals, nuts, and spices [27]. Some Aspergillus such as Aspergillus restrictus and Aspergillus penicillioides, are even more xerophilic, they grows very slowly or not at all at high aw (> 0.93) and are capable of growth down to at least 0.78 aw. Whereas, many penicillia can grow between 0.85 and 0.80 aw, but they compete poorly at these low aw levels unless other factors are in their favour. Still, a few Penicillium species such as P. brevicompactum and P.implicatum are ccapable of growth below 0.80 aw [30].”

Change:  Moreover, misquoted reference has been corrected. [31] “Agriopoulou, S.; Stamatelopoulou, E.; Varzakas, T. Advances in Occurrence, Importance, and Mycotoxin Control Strategies: Prevention and Detoxification in Foods. foods 2020, 137, 1–48”.

Comment 3: Line 110-112: The misconception about the production of the different aflatoxin types still remains. Aflatoxin M1 is the hydroxylated metabolite of aflatoxin B1 and can be found in milk or milk products obtained from livestock that have ingested contaminated feed. Aflatoxin M1 is not directly produced by molds, but by animals fed contaminated diets.

Response:  Thanks for your generous comment.  You are right, AFM1 is a hydroxylated form of AFB1and it is usually found in milk and milk products if the animals are fed with AFB1 contaminated feeds.  However, A.flavus and A.parasiticus are also able to produce AFM1 directly while higher aflatoxin contamination occured. Several researches have reported the detection of significant amounts of AFM1 from crops. I have cited two references (33; Line 232, 59) done in Ethiopia where AFM1 detected in crops as also depicted in Table 1 (Ref 33). Moreover, the concept has been supported by several previous studies cited in our manuscript (60, 61, 62). 

Change: More references which supported the concept are added and the change is highlighted in blue in the manuscript.

Example references:

  1. Chala, A.; Taye, W.; Ayalew, A.; Krska, R.; Sulyok, M.; Logrieco, A. Multimycotoxin analysis of sorghum (Sorghum bicolor L . Moench) and finger millet ( Eleusine coracana L . Garten) from Ethiopia. Food Control 2014, 45, 29–35.
  2. Getachew, A.; Chala, A.; Hofgaard, I.S.; Brurberg, M.B.; Sulyok, M.; Tronsmo, A.M. Multimycotoxin and fungal analysis of maize grains from south and southwestern Ethiopia. Food Addit. Contam. Part B Surveill. 2018, 11, 64–74, doi:10.1080/19393210.2017.1408698.

Supporting references cited

  1. Yabe, K.; Chihaya, N.; Hatabayashi, H.; Kito, M.; Hoshino, S.; Zeng, H.; Cai, J.; Nakajima, H. Production of M- / GM-group aflatoxins catalyzed by the OrdA enzyme in aflatoxin biosynthesis. Fungal Genet. Biol. 2012, 49, 744–754, doi:10.1016/j.fgb.2012.06.011.
  2. Abdallah, M.F.; Girgin, G.; Baydar, T. Occurrence of multiple mycotoxins and other fungal metabolites in animal feed and maize samples from Egypt using LC-MS/MS. Sci. food Agric. 2017, 97, 4419–4428, doi:10.1002/j.
  3. Sulyok, M.; Beed, F.; Boni, S.; Abass, A.; Mukunzi, A.; Krska, R. Quantitation of multiple mycotoxins and cyanogenic glucosides in cassava samples from Tanzania and Rwanda by an LC-MS / MS-based multi-toxin method. Food Addit. Contam. Part A 2014, 32, 37–41, doi:10.1080/19440049.2014.975752.

Comment 4: For coherency, all references to maximum levels in the manuscript should be made for either unprocessed products or for products for direct human consumption, but not a mixture of both. Authors should obtain latest consolidated version of COMMISSION REGULATION (EC) No 1881/2006 for proper quotations.

Response:  Thanks for your critical comment. Your comment has given us a very good chance to thoroughly revise the issue in this regard.

Change: We have gone through to the entire manuscript to use the term “… for products for direct human consumption…..”. The changes are shown in the manuscript highlighted with blue color. 

Comment 5: Line 173: the maximum level of fumonisins in maize is not 2000 ppb

Response: Thanks for your comment. The maximum level of fumonisins in maize intended for direct human consumption is about 1000 µg/kg.

Change: accordingly it has been corrected as ….. EU maximum limit (1000 µg/kg) set for other cereals like maize for direct human consumption [3].

Comment 6: Line 230: the maximum level of zearalenone in maize intended for direct human consumption is not 350 ppb

Response: Corrected accordingly asin maize intended for direct human consumption (100 µg/kg).

Change: The change shown in the manuscript line 230 in blue highlight

Comment 7: Table 1: typo in complimentary foods (correct complementary)

Response: Sorry for the spelling error committed. Corrected accordingly 

Comment 8: Line 336: the maximum level of aflatoxin B1 in processed cereal-based foods and baby foods for infants and young children is 0.1 ppb (COMMISSION REGULATION (EC) No 1881/2006 of 19 December 2006 setting maximum levels for certain contaminants in foodstuffs)

Response : The maximum limit for AFB1 is 0.1 accordingly as stated in the commission regulation “(COMMISSION REGULATION (EC) No 1881/2006 of 19 December 2006 setting maximum levels for certain contaminants in foodstuffs)”. However, the study we have discussed reported about the total aflatoxins (B1+B2+G1+G2) not AFB1. The maximum limits for the total AFs has not been treated in processed cereal-based foods and baby foods for infants and young children in the commission regulation mentioned. Thus we preferred to show the presence of higher AFs beyond the maximum limit (4 ppb) by referring “All cereals and all products derived from different cereals” as complementary foods are cereal based as well as this category of foods have a maximum limit for total aflatoxins.

Change: corrected as………. “total AFs exceeding the EU maximum limit (4 µg/kg) set for cereals intended for direct human consumption [4]”

Comment 9: Still uncorrected: EU establishes maximum levels for mycotoxins in foodstuffs (not recommended limits, permissible limits, etc). Revise terminology throughout manuscript.

Response: Thanks we have gone through the manuscript to make uniformly utilize the term “maximum limit”, according to your kind suggestions

Comment 10: Reference 49 is incomplete:

COMMISSION REGULATION (EC) No 1126/2007 of 28 September 2007 amending Regulation (EC) No 1881/2006 setting maximum levels for certain contaminants in foodstuffs as regards Fusarium toxins in maize and maize products

Response : Thanks for important comment; missing information completed  

Change :Reference (48) corrected as48. EC (European Commission). Commission regulation (EC) No 1126/2007 of 28 September 2007  amending Regulation (EC) No 1881/2006 setting maximum levels for certain contaminants in foodstuffs as regards Fusarium toxins in maize and maize products. Off. J. Eur. Union 2007, 1126, 14–17.”

Thanks again for reviewing our manuscript and your critical comments are very helpful to us !

Reviewer 2 Report

Very nice work responding to review comments.  I particularly appreciate the inclusion of the summary Table 1. 

Table 2 could still be improved by including data on fumonisins (since you mentioned in the text they are regulated), inclusion of units for the columns (presumably ppb, though that would change for fumonisin), and citing the references in a footnote.  But depending on Toxins's editorial policies that might require another round of review, which I think is not necessary.  Ideally this would be corrected editorially or in the proofing stage.   

Author Response

Reviewer two comments, author’s responses and manuscript changes

Thank you! We found your comments extremely helpful and have revised accordingly. Our responses are given in a point -by-point manner below. Changes to the manuscript are shown by a blue highlight.

Comments and Suggestions for Authors

Very nice work responding to review comments.  I particularly appreciate the inclusion of the summary Table 1. 

Table 2 could still be improved by including data on fumonisins (since you mentioned in the text they are regulated), inclusion of units for the columns (presumably ppb, though that would change for fumonisins), and citing the references in a footnote.  But depending on Toxins's editorial policies that might require another round of review, which I think is not necessary.  Ideally this would be corrected editorially or in the proofing stage.  

Response: Thanks for your generous comments. Table 2 has been improved. More regulated food items added in the table and according to your comment fumonisins standards with respect to regulated food items are depicted in the table 2.

Change: One column added in the table to include the fumonisins maximum limits. The modified table is shown below

Table 2. Maximum levels of mycotoxins (µg/kg) in Ethiopia

Commodities

Aflatoxins

Fumonisins (FUMs)

AFB1

Total AFs

AFM1

Peanut kernels

5

10

-

Peanut butter

4

10

-

Maize

5

10

-

Wheat

-

4

-

Sorghum

5

10

-

Barley

5

10

-

Teff flour

-

4

-

Injera

5

10

2000

Raw cow milk

-

-

0.05

-

Lentil

5

10

-

Chickpea

5

10

2000

Whole peas

5

10

2000

Split peas

5

10

2000

Dry beans

5

10

2000

Dry faba beans

5

10

2000

Kolo (roasted grain, ready-to-eat food) 

-

-

2000

Beer

5

10

-

Sources:  [26] and ESA (personal communications)

Thanks again for reviewing our manuscript. Your comments and suggestions were very helpful even for our future carrier as a professionals in the area 

Round 3

Reviewer 1 Report

The authors have reviewed the manuscript taking into account the comments and suggestions of the reviewers. 

In my view, the sentence about aflatoxin M1 (current lines 114-115) can be misleading. Although mycotoxin M1 could be directly synthesized by molds that are large producers of aflatoxins, this is not the pathway usually recognized. It is suggested to modify the phrase as follows:

Lines 114-115:
Fungi such as A. flavus and A. parasiticus are essential members of the genus Aspergillus, which produce different kinds of AFs, such as AFB1, B2, G1, and G2 [32], as well as AFM1 [33], which is the hydroxylated metabolite of AFB1 and is found in milk and dairy products obtained from livestock that have ingested contaminated feed.

Author Response

Reviewer one comments, author’s responses and manuscript changes

Thank you! We found your comments extremely helpful and have revised accordingly. Our responses are given in a point -by-point manner below. Changes to the manuscript are shown by a blue highlight.

Comments and Suggestions for Authors

The authors have reviewed the manuscript taking into account the comments and suggestions of the reviewers. 

Comment: In my view, the sentence about aflatoxin M1 (current lines 114-115) can be misleading. Although mycotoxin M1 could be directly synthesized by molds that are large producers of aflatoxins, this is not the pathway usually recognized. It is suggested to modify the phrase as follows:

Suggestion: Lines114-115: Fungi such as A. flavus and A. parasiticus are essential members of the genus Aspergillus, which produce different kinds of AFs, such as AFB1, B2, G1, and G2 [32], as well as AFM1 [33], which is the hydroxylated metabolite of AFB1 and is found in milk and dairy products obtained from livestock that have ingested contaminated feed.

Response: Thanks for your very important suggestions. We do share your concern, the sentence was a bit misleading. Presenting AFM1 as it is a hydroxylated form of AFB1 would make the concept more clear and understandable.

Change: The sentence has been modified accordingly and shown with a blue highlight in the manuscript.

Line 114-117 modified as follows “ Fungi such as A. flavus and A. parasiticus are essential members of the genus Aspergillus, which produce different kinds of AFs, such as AFB1, B2, G1, and G2 [32], as well as AFM1[33], which is the hydroxylated metabolite of AFB1 and is found in milk and dairy products obtained from livestock that have ingested contaminated feed.”

Thanks again for reviewing our manuscript. Your generous comments were very important even for our future carrier in the area.